# Research on Consumer Perception Regarding Traditional Food Products of Romania

**DOI:** 10.3390/foods12142723

**Published:** 2023-07-17

**Authors:** Ionica Soare, Constanta Laura Zugravu, Gheorghe Adrian Zugravu

**Affiliations:** 1Faculty of Economics and Business Administration, “Dunărea de Jos” University of Galati, Domneasca 47, 800008 Galati, Romania; ionica.soare@ugal.ro; 2Faculty of Engineering and Agronomy from Brăila, “Dunărea de Jos” University of Galati, Domneasca 47, 800008 Galati, Romania; 3Transfrontier Faculty, “Dunărea de Jos” University of Galati, Domneasca 47, 800008 Galati, Romania; adrian.zugravu@ugal.ro

**Keywords:** respondents, questionnaire, consumers, traditional food products

## Abstract

Traditional Romanian food products are an integral part of local culture and spirituality. These food traditions have been formed over the centuries and represent a particularly important part ofthe development of a circular economy in rural areas. In order to contribute to the development of this sector of activity we conducted a study on consumer perception of traditional Romanian food products. The purpose of this work was to identify the structure of the studies and the future directions of research related to the image of traditional food products through the bibliometric study, as well as the identification of consumer trends, of these products through a survey based on the questionnaire. The results obtained indicate that traditional Romanian food products are consumed by the majority of respondents participating in the survey and that this sector ofactivity has continuity in its development and presents a clear interest among citizens.

## 1. Introduction

The consumption of traditional Romanian food products supports the preservation of cultural heritage. In addition to promoting links with cultural heritage, traditional foods are important because they are often perceived as tastier, more nutritious, and less expensive than foods purchased from chain stores. Thus, by capitalizing on short supply chains, traditional foods are much more competitive than similar products sold by retailers. On the other hand, traditional products are obtained within the framework of local economy relations, thus providing the opportunity for rural development of the areas from which they originate. This research provides data for the foundation and development of nutritional policy as well as the component of policies regarding food security.

On the food market there is a growing offer of traditional products made available to consumers by producers, thus influencing their behavior, and generating trends in territorial development (European, national, local), research, and scientific development. The European authorities, including those from Romania, constantly monitor and regulate the agro-food market, ensuring these products are created according tocertification, labeling them to bear the mark of tradition of the cultural heritage of a local rural community, thus distinguishing them from other similar products belonging to the same category. These food products are distinguished mainly by their quality, respecting traditional production and/or processing methods and a recipe derived from traditional food patterns [1,2]. There are studies in this sense that indicate the connection of food cultures with health and well-being [2], this being performedby consolidating people’s sense of cultural identity. Perceived as authentic products, some consumers prefer these to table needles [3,4,5,6].

Culinary skills and practices, as part of the individual, collective, and territorial construction, differentiate the areas [7] and boost the attractiveness toconsumers and tourists. The market of traditional products in Romania has, 10 years after the implementation of the normative act of 2013 regarding the testing of traditional products, over 700 products thanks to a growing entrepreneurial network [3,8,9,10]. Rural areas have gradually come to be areas of concentration of traditional a products, such as Sibiu and Brașov, and show high consumer potential. The relationship between food consumption and consumer perception is close and significantly influences the decisions consumers make regarding their food [11,12]. Consumers’ perception of food is influenced by a variety of factors, such as the residential environment in which they were born and live in, the way of life chosen by each, food quality and safety, sensory characteristics, price, packaging, and nutrition [13,14,15]. These factors can determine consumers’ decisionsto buy, or not, a certain food and can directly affect the amount and type of food consumed [16,17,18].

The research performed in this article is a continuation of the research related to the buying behavior of local traditional food products. Thus, we sought to verify the classical hypotheses according to the socio-demographic characteristics of consumers and the price which influence the buying behavior oftraditional Romanian food products. We also analyzed the motivations and perceptions of consumers towards traditional Romanian food products.

The purpose of our research is to study the perception of consumers of traditional Romanian food products to identify consumption patterns and the factors that influence the purchase of these products. We aim to identify the existence of consumption habits that will support the development of the Romanian traditional food market. For this purpose, we chose a qualitative research methodology, based on a questionnaire among consumers, which allowed us to approach this topic in depth. The first part of this research consists of a review of the specialized literature through a double approach. In the first phase, we made a presentation of the supporting literature, which was followed in a second phase by a bibliometric analysis made with the help of the VOSviewer software. With the help of literature analysis, we were able to identify the main elements that can influence the decision-making process of the consumers of traditional Romanian food products, which will help us to better understand the purchasing behavior of these products.

## 2. Literature Review

Consumers’ interest in their own health has been the subject of several marketing studies. Currently, the agro-food system offers new products and food manufacturing technologies [16,17]. In the category of these innovations, due to their competitiveness, traditional food products are also present, owing to their nutritional characteristics, such as products with a low content of fat, sugar, salt, etc.; products enriched with fiber, omega 3 or 6, etc.; products without certain constituents salt, sugar, etc.; and dietary supplements. Understanding the in-depth analysis of consumer’s motivations and behavior is an important objective of the research in the context, especially as there is a lack of data on consumer perceptions of traditional Romanian food products.

Different works regarding the consumption behaviors of traditional food products emphasize the importance of the socio-demographics of consumer age, income, and educational level. Additionally, societal sensitivity is more important in adults aged between 36 and 65 than in young adults aged between 18 and 35, which shows a direct dependency relationship with age. At the same time, the sensitivity increases with the level of education [18,19].

Other works are oriented towards assessments of the identification of obstacles and motivations to consume traditional food products. French producers of local products reveal four main reasons to consume traditional local food products [20,21,22]:-reducing health risk;-rediscovering true flavors through the freshness of products;-commitment related to mental and social environmental concerns;-the search for the social bond that causes the consumer to interact with producers and other consumers.

The issue of price and willingness to pay is also widely addressed in this research on preferences for traditional food products. For certified traditional food products, consumers often accept higher prices than industrial ones and are willing to pay more to acquire them [23,24,25,26].

Behavioral variables are also related to eating skills, growing food, or cooking using traditional local products. Indeed, intentions to buy local traditional food products were significantly associated with growing food alone and preparing meals from traditional ingredients. The analysis of demographic characteristics in the specialized literature shows that they did not have a significant impact on the probability of buying local food, unlike behaviors associated with food knowledge or those related to the pleasure of cooking [27,28,29].

The subjective elements, related to the perception of traditional food products by consumers, are determining factors in the consumption decision. Among these, quality, in a broad sense, is an essential attribute. Along with quality, consumer perception highlights the role of factors such as freshness, taste, the importance given to supporting the local community, and environmental benefits [30,31,32].

Traceability is also a particularly important component in the consumption decision for the consumer of traditional food products. Consumers’ perceptions of traditional food products vary according to the frequency of purchase. The relationship between the frequency of purchase and the positive perception for traditional food products may mean that the number of consumers also determines an increase in the number of positive perceptions.

Gastronomy is directly related to agriculture, the production and consumption model, and the school contributes to the formation of behaviors, career orientation, and the thinking that reflects the realities of the economy and places [33,34,35,36,37]. As for the criteria used in choosing traditional products, they may vary from person to person, but may include the following:Authenticity—people may prefer authentic traditional products, which are made in a traditional way and use traditional ingredients and production methods [38,39,40,41].Quality—many consumers consider traditional products to be better in terms of quality, taste, and aroma than modern products [42,43,44,45,46].Source—some people prefer to buy traditional local or regional products because they think they are better than products from other regions or abroad [47,48,49].Environmental impact—many consumers are concerned about environmental impact and prefer traditional products that are sustainably produced and do not have a negative impact on the environment [50,51,52,53,54].

## 3. Methodology

### 3.1. Bibliometric Analysis Methodology

The main objective is to perform a research on the perception of traditional Romanian food products, starting from the realization of a bibliometric study on the main concepts approached in the field of traditional Romanian products and the identification of the main trends regarding the term traditional.Information extracted from the WOS database was used as scientific data, and the use of VOSviewer software provided the main analysis and reporting techniques. This paper presents a methodology for analyzing the scientific production of ISI-indexed works related to the marketing of traditional products, and it is believed that this could be applied in the assessment of the state of the art of any other field, by using the Thompson Reuters Web of Science, tools which allow the access and analysis information indexed in the main academic journals of 150 disciplines. This database is considered one of the main sources of information for bibliometric studies. The motivation of the present study is to investigate the evolution of scientific research based on the support policies of traditional products through review and quantitative bibliometric analysis. Correspondingly, this research aims to provide a critical overview of previous studies and identify the main challenges and opportunities for the implementation of policies to support traditional products. Traditional techniques for determining bibliographic contributions in a given discipline require the scientist to immerse themselves in the documents generated in their area of interest; however, this results in a time-consuming process of reading and analysis. In addition, results are generally subjective and difficult to reproduce. For this reason, and given the technological tools available today, it is recommended to use programs that facilitate the location of the material most relevant to the scientist.

Evaluation of the perception and preference of consumers on traditional Romanian food products is a complex topic that requires a quantitative approach based on powers and a qualitative approach based on linguistic terms. The quantitative approach to the analyzed topic is based on mathematical tools for interpreting the results. Quantitative bibliometric analysis is used in many fields and is based on the premise that a scientific work acquires a greater value only when it is analyzed by the scientific environment. The most important criterion used in the case of researching the perception and preference of consumers on traditional Romanian food products refers to the indexing of the research in Web of Science databases. We used this selection criterion in the bibliometric research, because an article indexed in Web of Science goes through an evaluation procedure and receives recognition in the field by citing it by other authors. The present bibliometric analysis by its quantitative nature managed to provide consistent results related to the main concepts encountered regarding the perception and preferences of traditional Romanian food products which were then verified through a questionnaire-based analysis among the consumers of these products.

The main objective of this bibliometric study into the perception of traditional products is to demonstrate the necessity of our study in the context of the lack of scientific research on this topic in Romania. The research of the database of ISI indexed articles, related to the perception of traditional products, was performed for ten years (2012–2022) on the trends regarding the traditional product concept. The literature was extracted and analyzed using the Web of Science database. VOSViewer software was used to identify and visualize key trends, influential authors, and journals. The 279 WOS articles from 2012 to 2022 were selected based on three main criteria which were:-Topics regarding traditional Romanian food products.-Document type “article”.-Year of publication in the period 2012–2022.

Data analysis, with bibliometric techniques, is especially useful when dealing with large amounts of information, such as the literature production of a particular discipline. This paper is limited to a study of 279 WOS articles from English-language journals addressing the topic of “Romanian traditional products” for the period from 2012 to 2022 and aims to present an overview of what has happened in the period from 2012 to 2022 in this field. We will analyze the evidence from the database available on the Web of Knowledge to determine which journals have the highest impact, we will present an overview of how the number of publications has evolved in the area, as well as the relationship between the most important authors through co-citation analysis and the authors with the largest production of articles, and investigate the connection between “traditional Romanian products” through a correlation study of the most used keywords.

The works, since then, are mainly case studies which, by their very specific nature, have been difficult to apply to other types of organizations. In this period there are two main perspectives, one focused on the process of generating and implementing strategies and the other focused on understanding the relationship between strategy and performance. It is possible that the most frequently cited papers have exerted a greater influence on the discipline than the least cited ones given that the more often a given paper is cited, the greater its influence in the scientific development of the analyzed field. Therefore, the criteria we use to define which elements, publications, and authors have had the greatest impact on the development of the “Romanian traditional products” field is the number of citations, and so an article with more citations will be considered more important than an article with less. Co-citation is a co-occurrence link and occurs when two literary elements such as articles or authors which are cited by a third party. Therefore, it is expected that with a higher frequency of citing, there is a greater affinity between the elements. Activity values provide information about the quantity and impact of scientific production, indicators such as the impact factor, or the numbers of articles are examples of this type of metric. On the other hand, the metrics or relational indicators allow us to know how a discipline is constituted. The study co-cites, also called the first generation, and the relationship of the associated words, also called second generation, are the two types of analysis that allow reveal the relational indicators. The bibliometric study based on 279 articles obtained from the Web of Knowledge, and published in ISI indexed journals, are processed with the VOSviewer software by generating matrices and relational maps. For the study of the main authors, we first present the 50 researchers who published the most articles, then with the analysis of co-citations we can observe the relationship between the authors. Regardless of the year of publication, the VOSviewer tool allows interpretation of the results of these relationships, grouping into segments those elements that they are tied, such associations, by author, by group, where I can see group results, the number of occurrences, and the most important authors for each group.

### 3.2. Consumer Survey and Data Analysis Methodology

Key words from the four clusters in the bibliometric study and the specialized literature [6,55,56,57,58], constituted the basis for establishing and formulating the items. Additionally, the authors’ discussions with friends, colleagues and relatives, the authors’ critical thinking as consumers of traditional products, as well as the “National Register of Traditional Products” (RNPT) from Romania represented ideas in the way of choosing the items.

The data for the research were collected using a survey based on a questionnaire. The research was performed among residents from the south-east development region of Romania using a convenience sampling method. Using the filter question “do you consume traditional products?”, the criteria for inclusion and exclusion of study participants was established. Thus, by answering “yes” the representative answers were selected (89.8% of the total), the rest not being interested in traditional products; some of the uninterested (9.2%) did not rule out the possibility of being interested in the future.

Thus, of the 303 forms completed online, based on social networks: Facebook and Whatsapp, between January and April 2023, 272 were validated in the end. The questionnaire includes 2 parts. The first part includes socio-demographic characteristics (gender, age, place of birth, current residence, studies completed, and residential environment).

The 2nd part includes a number of 20 items (ST1–ST20) regarding the importance of motivations/preferences/criteria of consumers of traditional food products and 3 other multiple-choice questions related to travel, preference compared to another category of food products, and food education.

For the 20 items, the respondents evaluated on a Likert Scale the importance of different preferences and criteria in the decision to choose and consume traditional products. Likert Scale results were scored as follows: 5 = very important/very satisfied; 4 = important/satisfied; 3 = indifferent (neutral); 2 = less important/slightly satisfied, and1 = a little important/dissatisfied. Methods of descriptive and inferential statistics were used for analysis of the collected data. The normality of the data was checked using Shapiro–Wilk test. The results show that all significance levels are lower than 0.05, indicating that all variables used in this study present non-normal distribution. Thus, the non-parametric Mann–Whitney and Kruskal–Wallis tests were used to analyze if there were significant differences regarding the residence consumers’ perception of the motivational factors in the consumption of traditional products based on socio-demographic characteristics. Data were analyzed using SPSS Statistics for Windows, version 23.0.

## 4. Results

### 4.1. Results on Bibliometric Analysis

We performed several types of analyses regarding the perception of Romanian traditional food products, we analyzed the period of 2012–2022 in its entirety, after which we analyzed the most cited articles and the most recently accessed articles. The countries, from which the articles with the greatest contribution in the field of traditional Romanian products come from, are presented in Table 1 and Figure 1.

Based on the linking of the bibliographic references of the 279 WOS articles analyzed, a map of the evolution of the relationships generated by the development of the concept of traditional Romanian products in the last 10 years (2012–2022) was obtained. The map uses the rainbow color code, the red color being assigned to the newest relationships (Figure 1).

According to the analysis of co-citations within the analyzed articles in the field of traditional Romanian products, a map of their density was obtained on the main WOS indexed journals, corresponding to the intensity of the color density (Figure 2).

Research themes were extracted and analyzed to identify and visualize main trends, (influential) authors and related journals. There are five major keyword groups related to the database related to traditional Romanian products in the WOS articles from 2012 to 2022, which we determined based on four thematic clusters which are corresponding to the intensity of the color density (Figure 3 and Figure 4):-Cluster 1: adulteration, antioxidant, antioxidant activity, identification, in vitro, products, and quality, with red in Figure 4.-Cluster 2: food, innovation, Romania, technology, and trends, with green in Figure 4.-Cluster 3: consumers, impact, and internet, with blue in Figure 4.-Cluster 4: behavior, health, and risk, with yellow in Figure 4.

The relationships between the terms identified in the 4 clustered a dynamic over time that is reflected through the trends of the last 5 years through the spectrum of rainbow colors from blue to red (the newest trends, according to Figure 5).

Thus, identifying significant trends is vital from a long-term perspective. The optimal way to reach a real overview is to provide a complex visualization of the literature through bibliometric networks. This idea is not original, because there have been similar efforts in the society of scientists until now, but the essence of identifying bibliometric networks. According to this view, we can state that there are these trends in the scope of scientific literature on traditional product marketing, and the cultural approach should be one of the identified trends. Therefore, the cultural approach is not missing from the list of trends identified as it is only a very recent trend that does not have enough support in the scientific literature so far to be considered an autonomous trend and nowadays, it is partially included in the broader topics. From this bibliometric analysis, the presence of new technologies and approaches related to traditional products is noted. Regarding the main problems addressed in the field of Romanian editorial products during the period of this study, we observe what these are and their interrelation in the following graphs based on the analysis of keywords; the red areas are where the greatest impact is concentrated, in addition to the corresponding correlation matrix.

### 4.2. Results on Consumer Survey

According to the attached table (Table 2), the situation of the Romanian respondents is: the majority is owned by women, in the age groups 18–19 years, respectively, 20–29 years; the high percentage of women who have their place of birth and current residence in the urban environment, as well as those with high school education; two groups according to the residential environments covered—those from urban and rural areas (rural + urban and rural)—results in equal proportions (50% each).

The respondents evaluated on a Likert Scale the importance of different preferences and criteria in the decision to choose and consume traditional products. The analysis shows high importance scores with a mean > 4 (Table 3).

For the score attributed to “custom passed on by parents and grandparents” (ST1), the explanation given is related to the eating habits that are formed in the family, where the contact with this is maintained closely until the age of completing high school studies, even university ones. Other motivations with a high score are: “food safety” (ST5), “health benefits” (ST6), “expiry date” (ST9), and “price” (ST10).

The ability to perceive information about the quality of food products differs from one consumer to another [59], therefore price is not the only quality indicator that would guide consumers’ purchase decisions. More subjective elements, related to consumer perception, thus represent factors determined in the consumption of traditional food products where quality is an essential element of competitiveness in relation to food industry products. On the one hand there are works that highlight the role of perception for traditional food consumers [60,61], and on the other hand we find works that point out that the analysis of these motivations and perceptions is never related to the purchase of local traditional food products [62,63,64].

The orientation of the population towards “food security” and “health benefits” can be explained by the current situation and structure of the food market and the former COVID-19 pandemic, which is partially or totally reflected in the other preferences and stated criteria, and takes into account the logic of the meaning/content of each of them. Statistically significant values for *p*-values are obtained for certain socio-demographic characteristics (Table 4).

We notice a greater openness of the age groups 18–19 years, respectively, 20–29 years, regarding the choice of traditional products based on the motivation “curiosity” (ST4, *p* = 0.002, Table 3). The same criterion—“curiosity”—is preferred by respondents who have completed high school studies (*p =* 0.002, score 3.83) and those in the 18–19 age group; this result is not necessarily related to the level of training of the respondents, because those with completed university studies also have a fairly high degree of curiosity (grade: 3.41). Additionally, respondents under 30 years old are open to choosing traditional products through the statements “recent experience” (ST2, *p =* 0.042, score 3.42 for 18–19 year olds) and “friend recommendation” (ST3, *p =* 0.033, score 3.38 for the 18–19 year old age group and 3.22 for the aged 20–29).

The data confirm for Romania similarities with situations in other Eastern European countries. For example, a study on consumer behavior in Poland highlights statistically significant values for the “curiosity” criterion in food consumption as being very closely related to the tendency to consume these products, and the “recommend to a friend” criterion [55,65,66,67,68]. In another study on the attitudes of young adult consumers (18–30 years) from seven European countries (Greece, Bulgaria, Romania, Slovenia, Croatia, Denmark the marquis of France) [69,70,71,72,73,74], it is reported that young people and adults from eastern European countries are more easily influenced by their friends and close relatives compared to those from western countries [31,56,75,76,77,78,79,80]. The criteria “curiosity” (ST4) and “recommendation to a friend” (ST3) are statistically significant in the “residence” group (*p* < 0.05), with those who live in the urban environment being more open in all four selection criteria for traditional food products Additionally, the criterion “label” (ST7) (*p =* 0.010 < 0.05) is statistically significant for the socio-demographic characteristic “place of birth”, where high values are attributed to respondents born in rural areas with rural origins. It seems that those born in the rural environment want a “certification” for the fact that the purchased product conforms to the “taste of childhood”, i.e., it is authentic. Familiarity with the label leads to knowledge about these products in relation to other types of food, as shown in the study from Slovakia [56,81]. Although those who live in the countryside and who lived in a residential environment close to rural may be attached and more selective compared to traditional products, the results of the two groups of consumers according to the residential environments traveled—only urban and rural (rural + urban and rural)—in the case of statements ST1–ST10 do not indicate significant statistical differences with consumers whose residential environment is only urban, only rural, but also urban and rural. The price (ST10) does not discourage consumers; the quality of traditional food products satisfies preferences.

The analysis, according to the Likert Scale, shows scores of high importance (average > 4) for all the characteristics of the established traditional products (Table 5). In principle, color, taste, smell, external and internal appearances are the sensory characteristics appreciated differently from person to person. Of all the sensory characteristics of the traditional food, “color” (ST11) registers more statistically significant values on the socio-demographic indicators: gender (*p =* 0.011), women being the ones who appreciate more; origin (*p =* 0.05) and the residential environments visited (*p =* 0.004) with high marks from the respondents who lived and live in connection with the countryside (Table 3). It seems that this category of respondents, who are closer to nature, appreciate the quality and freshness of a product better by its color, and also, “external aspect” (ST14) all theirs is attributed (*p =* 0.015—rural or urban environments only).The characteristic “external appearance” (ST15) also has significant values in the gender component, like color, where the highest score is also given by women. Related to visual quality, appearance, like color, is another main reason for purchasing and consuming products, as indicated by studies for some countries/areas—Nova Scotia [60], EU countries (Belgium, France, Italy, Norway, Poland and Spain) [31,82,83,84,85,86]. We note that the female population is more focused on selecting and buying traditional products, which indicates that women still maintain the major role in the family in terms of food consumption and, at the same time, maintaining the traditional family model in Romania, as well as in other countries, such as Poland, the Republic of Moldova. Color and appearance are the characteristics that influence the behavior of Romanian respondents as consumers [87]. However, the respondents who lived and still live in the rural environment, according to the results of our study, remain encouraged in the patterns of traditional food behavior, as indicated in the study) [55,88,89,90,91,92,93,94].Traditional products do not meet their expectation, which leads them to resort to re-purchasing the products and/or directing others to them purchase. The results from the sensory evaluation correlate with the label (ST7), which confirms that traditional food fits this type of consumer. The label for traditional products is an indicator of sensory evaluation, as it is well known the fact that the information on the label, regardless of the product, significantly affects the sensory judgment of the consumer. The results obtained from the company study also reflect the characterization of the traditional product. The perceptions of the sensory characteristics are related not only to the intrinsic characteristic, but also to the raw material and processing technology [95,96].

Among the five categories of traditional products (Table 3), meat and/or meat products, milk and/or dairy products, fruits/vegetables, and bread/bakery and pastry products indicate high importance scores (average > 4), these being part of the category of basic food products. Statistically significant values are for milk and/or dairy products (ST17) on the gender component (*p =* 0.029; score 4.35 women), for bread/bakery and pastry products (ST19) on the main socio-demographic characteristics (*p =* 0.034, score4.39 for the age group 18–19 years) and studies (*p =* 0.031, with high marks for post-secondary and high school studies, where those in the 18–19 year group fall).

The request from the questionnaire regarding the main reasons for choosing a holiday destination in Romania (the Carpathians and the adjacent area/Danube Delta, Littoral) was based on checking whether the respondents on their preferences to consume traditional and local food products, to test products and to introduce them to the diet later. This was also the reason “local cuisine/traditional cuisine” was chosen by 48.9% of 272 respondents, occupying the 3rd position percentage, after natural tourist resources (74.6%) and hospitality (56.6%). As it emerges from the research, this situation suggests, at the level of small producers of traditional and local products in Romania, the need to approach new ways in consolidating the position on the market by establishing a stronger brand, through stories, offering touristic experience, thus putting more consumers in contact with their products. In the case of the question if in the future the respondents prefer the traditional product instead of the “fast food” type product, 68% of them answered with “yes” and 23.5% answered with “I do not know”. So, consumers are oriented towards a healthy diet and young people and adults are more aware, more informed, more educated.

Studies show that fast food consumption is more popular among children and adolescents, such as in Turkey [97] and Pakistan [98]. Since a significant percentage of the respondents are between 18 and 29 years old, some of whom are still in the process of completing their studies, responded to the question “do you think a gastronomic education is necessary in the context of adaptation to change (generated by the climate, economic crises, health), which is the specific local calculation?” answered “yes”(85.7% Of the respondents)answered and 9.6% answered “I do not know”. The high percentage of discussions with respondents in the 18–19 and 20–29 age groups on this idea derives from their desire to have more knowledge about the consumption of foods with super health benefits.

## 5. Conclusions

Currently there is a presence of new technologies in scientific approaches regarding traditional food products, as indicated by the bibliometric study. This bibliometric analysis represents a method of quantitative analysis of the most important concepts found in the scientific research performed in the field of traditional Romanian food products, which were grouped into four clusters. From the perspective of the frequency of these key terms that were found in the bibliometric research, it is noted that in the last 3 years there was a tendency to focus research in the field of traditional Romanian food products on the positive impact of the consumption of these products, on the health of the consumer by reducing the food risk, and the qualitative contribution provided by antioxidants. All these results constitute the support of future research regarding the development of nutritional policies as the basic orientation of food security.

The novelty and diversity of the products on the food market make the consumer’s behavior continuously dynamic. Traditional food products offer the opportunity to implement innovative businesses based on years of empirical research.

The research on the perception of consumers of traditional Romanian food products made it possible to highlight the purchasing behavior of these products based on a positive perception, in terms of individual benefits (quality, taste, and freshness) and on the significant impact of the factors that determine consumption habits(age, level of education, and place of residence).

Respondents, especially young adults (18–29 years), are more open to new food and are aware of the quality and origin of food in the food market. This situation can be reflected in the increase in the competitiveness of producers of traditional products. Additionally, the respondents, regardless of age and gender, emphasized in the purchase and consumption decision sensors that color and appearance (interior and exterior) that women are more attentive to color and interior appearance than men. Those who live in the countryside and who have lived and still live in the rural environment appreciate the color and the exterior appearance more; they are still ingrained in the patterns of traditional food behavior. In the diversity of the market and according to gender and age, preferences vary in the case of consumption of groups of traditional products; women appreciate milk and/or dairy products more, and bread/bakery and pastry products are preferred by young adults. In addition to that mentioned, the respondents prefer traditional cuisine, in the future they want to remain attached to these traditional products and recommend resorting to a gastronomic education in school.

The main limitation of the study is the non-probabilistic method used for the collection of data. Convenience sampling was applied in the present study, the questionnaire being mainly distributed among students, while the remaining was distributed randomly through social avenues. Another limitation of the study is that it was focused on a single development region of the country. For these reasons, the results obtained in this study may be considered as starting points for future studies rather than extending them to the entire country territory. The limits of this research also refer to the fact that the respondents are not experts in the sensory analysis of food products and the analysis by them of some specific characteristics is particularly difficult.

## Figures and Tables

**Figure 1 foods-12-02723-f001:**
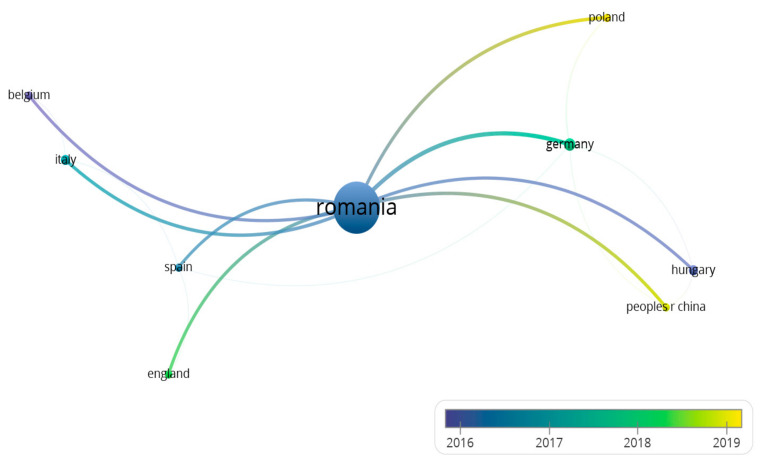
Visualization of the evolution of relations by country, generated by the development of the concept of traditional products in the last 10 years (2012–2022).

**Figure 2 foods-12-02723-f002:**
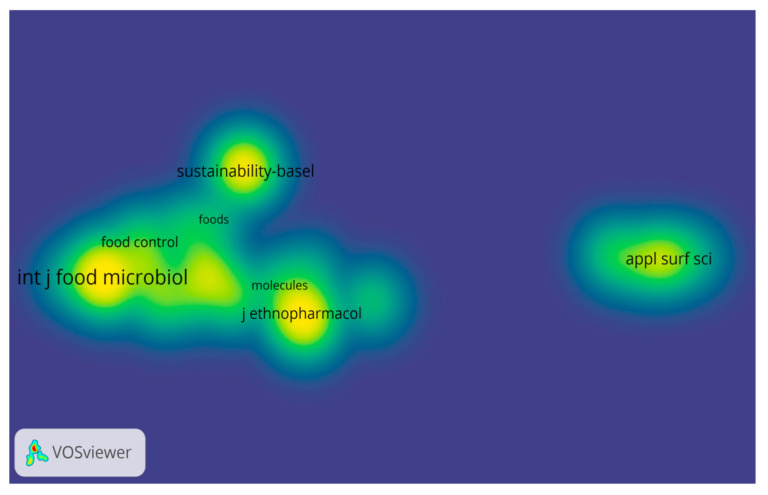
Visualization of the density of co-citations related to traditional Romanian products.

**Figure 3 foods-12-02723-f003:**
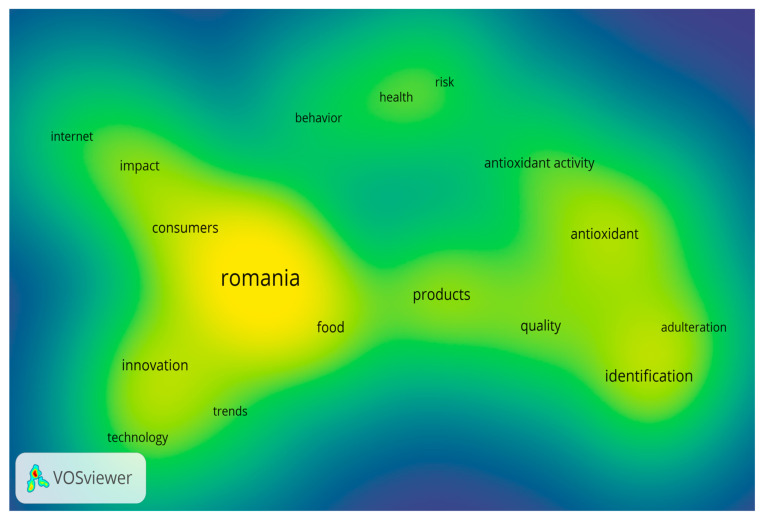
Viewing the density of terms from the 4 identified clusters.

**Figure 4 foods-12-02723-f004:**
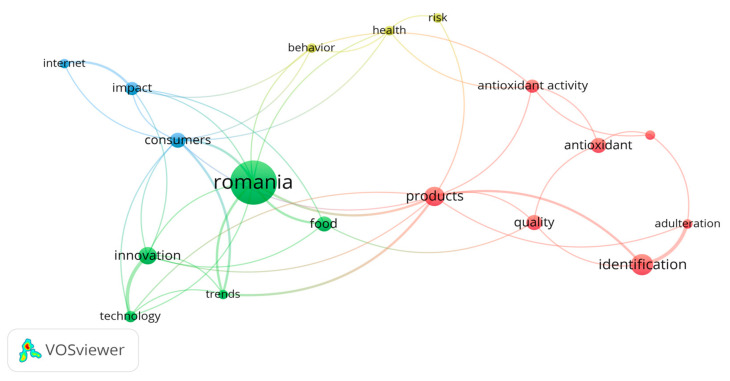
Visualization of the relationships between the terms of the 4 clusters.

**Figure 5 foods-12-02723-f005:**
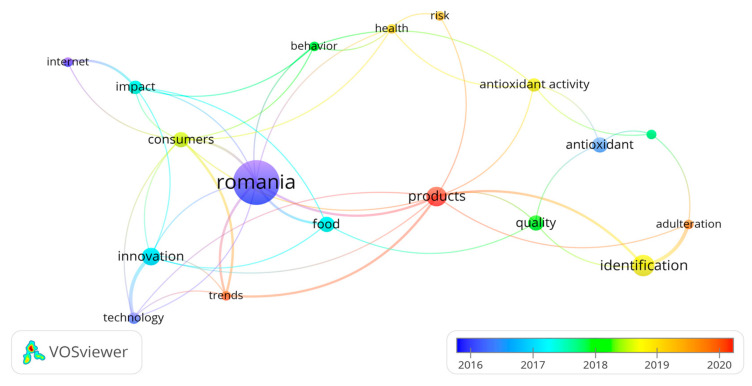
Viewing the current trends of the relationships between the terms of the 4 clusters.

**Table 1 foods-12-02723-t001:** The contribution by country to the development of the concept of traditional Romanian products.

Country	Documents	Citations	Total Link Strength
Romania	229	1355	41
Belgium	5	95	5
England	5	99	6
Germany	11	109	15
Hungary	7	129	6
Italy	7	72	7
China	5	61	7
Poland	5	28	8
Spain	5	174	7

**Table 2 foods-12-02723-t002:** Socio-demographic characteristics of respondents.

Characteristic	Category	Respondents	%
Gender	Female	181	66.5
	Male	91	33.5
Age	18–19 years old	127	46.7
	20–29 years old	60	22.1
	30–49 years old	42	15.4
	50–70 years old	43	15.8
Place of birth	Urban	167	61.4
	Rural	105	38.6
Current residence	Urban	216	79.4
	Rural	56	20.6
Completed studies	high-school studies	181	66.5
	post-High school studies	9	3.3
	University studies	82	30.1
Residential environments covered	You come from the city	136	50.0
	Rural = only rural +(rural + urban)	136 (of which 90 in rural areas + 46 in both urban and rural areas)	50.0

**Table 3 foods-12-02723-t003:** The importance of motivations, preferences and criteria in the choice of traditional products.

Statements		Scale (%)						
		1	2	3	4	5	MEAN	SD
Customs handed down by parents and grandparents (ST1)		3.31	6.62	9.56	39.71	40.81	4.08	1.28
Recent Skill (ST2)		13.97	9.93	30.88	35.66	9.56	3.17	1.18
Recommandation of a friend (ST3)		12.87	12.87	26.47	40.07	7.72	3.17	1.16
Curiosity, trying new food (ST4)		7.72	8.46	13.60	50.74	19.49	3.66	1.16
Food safety (ST5)		1.10	2.57	6.25	32.72	57.35	4.43	1.37
Health benefits (ST6)		1.47	1.47	10.66	27.57	58.82	4.41	1.37
Label (ST7)		8.46	5.15	16.18	40.07	30.15	3.78	1.22
Label atractivity (ST8)		13.24	8.09	20.96	35.66	22.06	3.45	1.44
Expiration date (ST9)		2.57	0.37	4.04	13.60	79.41	4.67	0.98
Price (ST10)		4.41	4.04	11.03	36.76	43.75	4.11	1.05
Sensory characteristics	Color (ST11)	1.5	1.5	8.5	37.1	51.5	4.36	0.81
	Taste (ST12)	0.37	0.37	1.47	11.40	86.40	4.83	0.68
	Smell (ST13)	0.4	0.7	1.5	20.2	77.2	4.73	0.67
	External appearance (ST14)	0.74	1.10	5.51	30.88	61.76	4.52	0.74
	Internal appearance (ST15)	0.74	0.74	4.04	24.26	70.22	4.63	0.72
Product groups	Meat and/or meat products (ST16)	1.5	4.4	7.0	33.1	54.0	4.34	0.90
	Milk/or milk products (ST17)	1.47	4.41	6.99	33.09	54.04	4.34	0.90
	Vegetables/fruits (ST18)	1.1	2.2	9.9	39.7	47.1	4.29	0.82
	Bread/bakery and pastriesproducts (ST19)	1.10	4.41	12.50	36.76	45.22	4.21	0.91
	Fish/or fish products (ST20)	3.31	6.25	18.75	34.19	37.50	3.96	1.12

**Table 4 foods-12-02723-t004:** Results of the Mann–Whitney U test and the Kruskal–Wallis test of socio-demographic characteristics and preferences and criteria in choosing traditional products.

Characteristic		Category	ST1	ST2	ST3	ST4	ST5	ST6	ST7	ST8	ST9	ST10
Gender	Mann–Whitney	Female	4.14	3.14	3.19	3.64	4.46	4.45	3.85	3.44	4.76	4.12
		Male	3.97	3.23	3.12	3.69	4.35	4.32	3.66	3.47	4.48	4.11
		*p*-value	0.271	0.418	0.674	0.772	0.363	0.465	0.153	0.984	0.174	0.711
Age	Krushal-Wallis	18–19	4.08	3.42	3.38	3.94	4.46	4.39	3.87	3.65	4.76	4.17
		20–30	3.98	3.00	3.22	3.75	4.42	4.18	3.55	3.25	4.60	4.00
		30–50	4.05	2.98	2.76	3.19	4.33	4.69	3.98	3.43	4.60	4.26
		50–70	4.26	2.86	2.88	3.16	4.44	4.49	3.67	3.16	4.56	3.98
		*p*-value	0.547	**0.042**	**0.033**	**0.002**	0.973	0.103	0.275	0.150	0.410	0.663
Place of birth	Mann–Whitney	Urban	3.99	3.07	3.13	3.60	4.46	4.40	3.66	3.41	4.61	4.02
		Rural	4.23	3.33	3.24	3.74	4.38	4.42	3.98	3.52	4.76	4.26
		*p*-value	0.067	0.057	0.617	0.459	0.589	0.826	**0.010**	0.358	0.313	0.085
Current residence	Mann–Whitney	Urban	4.07	3.24	3.26	3.75	4.43	4.39	3.80	3.48	4.67	4.10
		Rural	4.13	2.89	2.82	3.30	4.43	4.48	3.71	3.34	4.66	4.18
		*p*-value	0.472	0.057	**0.024**	**0.023**	0.873	0.407	0.646	0.582	0.936	0.174
Studies	Krushal-Wallis	high-school	4.09	3.28	3.28	3.83	4.50	4.41	3.83	3.54	4.73	4.19
		post-high school	4.22	2.78	2.33	2.44	3.67	4.67	3.78	3.67	4.89	4.00
		university	4.05	2.96	3.01	3.41	4.35	4.38	3.67	3.23	4.51	3.95
		*p*-value	0.722	0.144	0.055	**0.002**	0.053	0.506	0.800	0.217	0.166	0.081
Residential environments covered	Mann–Whitney	Only urban	4.00	3.07	3.18	3.65	4.40	4.31	3.72	3.46	4.60	4.09
		Rural = only rural + (rural and urban)	4.16	3.27	3.16	3.67	4.46	4.51	3.85	3.44	4.74	4.14
		*p*-value	0.289	0.142	0.667	0.873	0.631	0.084	0.184	0.810	0.162	0.352

Note: *p *< 0.05. *p *< 0.01.

**Table 5 foods-12-02723-t005:** Results of the Mann–Whitney U test and the Kruskal–Wallis test of socio-demographic characteristics and preferences and criteria in choosing traditional products.

Characteristic		Category	ST11	ST12	ST13	ST14	ST15	ST16	ST17	ST18	ST19	ST20
Gender	Mann–Whitney	Female	4.44	4.86	4.77	4.56	4.69	4.29	4.35	4.34	4.28	3.93
		Male	4.19	4.78	4.66	4.43	4.49	4.44	4.15	4.21	4.07	4.02
		*p*-value	0.011	0.412	0.180	0.075	0.027	0.303	0.029	0.211	0.095	0.509
Age	Krushal-Wallis	18–19	4.30	4.86	4.75	4.57	4.68	4.49	4.41	4.36	4.39	3.89
		20–30	4.40	4.92	4.77	4.53	4.58	4.25	4.20	4.23	4.15	3.78
		30–50	4.48	4.76	4.67	4.57	4.67	4.14	4.19	4.29	3.88	4.05
		50–70	4.35	4.70	4.70	4.30	4.49	4.21	4.14	4.19	4.07	4.35
		*p*-value	0.626	0.525	0.953	0.142	0.249	0.167	0.284	0.644	0.034	0.082
Place of birth	Mann–Whitney	Urban	4.26	4.81	4.69	4.47	4.56	4.37	4.26	4.26	4.24	3.94
		Rural	4.50	4.87	4.80	4.59	4.72	4.29	4.32	4.34	4.15	4.00
		*p*-value	0.05	0.757	0.363	0.358	0.190	0.358	0.682	0.390	0.384	0.749
Curent residence	Mann–Whitney	Urban	4.34	4.82	4.72	4.53	4.62	4.40	4.31	4.29	4.23	3.95
		Rural	4.43	4.86	4.77	4.48	4.66	4.11	4.20	4.32	4.13	4.00
		*p*-value	0.441	0.682	0.638	0.928	0.674	0.174	0.289	0.624	0.904	0.697
Studies	Krushal-Wallis	High-school	4.35	4.86	4.75	4.55	4.67	4.38	4.34	4.35	4.30	3.91
		Post-high school	4.67	5.00	5.00	4.67	4.89	4.44	4.56	4.33	4.44	4.56
		University	4.34	4.76	4.66	4.44	4.50	4.24	4.15	4.16	3.98	4.02
		*p*-value	0.563	0.554	0.283	0.402	0.092	0.547	0.316	0.229	0.031	0.220
Residential environments covered	Mann–Whitney	Only urban	4.21	4.83	4.68	4.41	4.57	4.49	4.33	4.21	4.21	4.03
		Rural = only rural + (rural + urban)	4.50	4.83	4.79	4.63	4.68	4.19	4.24	4.38	4.21	3.90
		*p*-value	0.004	0.764	0.099	0.015	0.089	0.085	0.638	0.142	0.787	0.401

Note: *p *< 0.05. *p *< 0.01.

## Data Availability

Data is contained within the article.

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
