# Peer review of "Research on Consumer Perception Regarding Traditional Food Products of Romania"

_foods, 2023, doi:10.3390/foods12142723_

Round 1
Reviewer 1 Report
Dear authors
General comments:
the paper about Consumer Perception Regarding Traditional Food Products of Romania is suitable for the journal Foods.
The authors have shown a good knowledge about the topic and sound scientific methodology use.
Data were suitably used and proccessed and the results are presented in a sound scientific manner.
Specific comments for major changes to be done:
The only quest for smaller intervention is in the introduction part where tourists and tourism were described, while it has no actual importance for the topic of the paper, neither to the concept of Romanian trditional food.
If the authors insist on the tourists topic then 2 paragraphs should be separated
1. for Romanian native people - domestic market
2. for external people - tourist market
then descibe for each market their consumer behaviour toward traditional Romanian food.
The same principle should then be used in literature review and results and conclusions.
You have also mentioned Croatia as example of food research, but none of the papers were cited for Croatian cases ?? Neither in Table 1.
Technically some errors were found:
1.the pictures should be enlarged, they are now very small and unreadable
2. references should be checked e.i. year in bold
3. Table 1 is in colour , it shld be in plane text without border lines
best regards
the reviewer.
Author Response
I took your recommendation into account and gave up the approach related to tourism.
I removed from the abstract, introduction, results and conclusions the paragraphs related to tourism that are not important for the topic of the paper.
The references have been expanded to include works related to Croatia
The figures have been put in an enlarged format
The references have been modified with year in bold
The Table 1 has been replaced without border lines and color

Reviewer 2 Report
The article entitled "Research on consumer perception of traditional Romanian food products" presents information on the perception of Romanian consumers on traditional food products. However, there are several aspects that make the manuscript very weak.
What is the purpose of the bibliometric analysis and the survey on the preference of traditional foods among consumers? I see a gap in both analyses.
Section 2 of the manuscript is not a literature review of the topic, rather it is a methodological level description of a review of the identified scientific production, which should be included in the methodology of the article and described in the results.
Section 2 is not a literature review, it is a bibliometric analysis, therefore it should go in the results section.
The objective of the research is not clear; the authors should state it emphatically in the introduction of the article.
In addition, the authors should conduct a proper literature review of the central theme of the manuscript: traditional foods and consumer perception and preference.
Authors should justify with sources the criteria on which they based their bibliometric analysis.
The authors should review the quality of the figures; figures 3 and 4 are of low quality.
It is necessary to establish the criteria for inclusion and exclusion of study participants. Likewise, it is necessary to justify the design of the questionnaire used, how the questionnaire items were determined.
The analysis of statement 19 with respect to gender is greater than 0.05, the authors should review this detail in Table 5.
The authors should discuss their findings according to the literature on the perception and consumer behavior of traditional foods.
I believe that the authors are excluding key sources on the perception and motives for consumption of traditional foods. Below is a list of important sources:
Guerrero, L., Claret, A., Verbeke, W., Enderli, G., Zakowska-Biemans, S., Vanhonacker, F., Issanchou, S., Sajdakowska, M., Granli, B. S., Scalvedi, L., Contel, M., & Hersleth, M. (2010). Perception of traditional food products in six European regions using free word association. Food Quality and Preference, 21(2), 225–233. https://doi.org/10.1016/j.foodqual.2009.06.003
Renko, S., Cerjak, M., Haas, R., Brunner, F., & Tomic, M. (2014). What motivates consumers to buy traditional food products? Evidence from Croatia and Austria using word association and laddering interviews. British Food Journal, 116, 1726–1747. https://doi.org/10.1108/BFJ-02-2014-0090
Serrano-Cruz, M. R., Espinoza-Ortega, A., Sepúlveda, W. S., Vizcarra-Bordi, I., & Thome-Ortiz, H. (2018). Factors associated with the consumption of traditional foods in central Mexico. British Food Journal, 120, 2695–2709. https://doi.org/10.1108/BFJ11-2017-0663
Wang, O., Gellynck, X., & Verbeke, W. (2016). Perceptions of Chinese traditional food and European food among Chinese consumers. British Food Journal, 118, 2855–2872. https://doi.org/10.1108/BFJ-05-2016-0180
Proofreading of the manuscript should be carried out.
Author Response
Section 2 was moved to the methodology and described in the results.
In the introduction section the objectives of this research have been clear described.
The literature review section has been added.
In the methodology section, I introduced a paragraph in which we presented the criteria for inclusion and exclusion of study participants, and also justified with sources the criteria on which our bibliometric analysis is based.
The quality of figures 3 and 4 has been increased.
In the section methodology we added a paragraph were justified the design of the questionnaire used
The statement 19 regarding gender have been corrected.
In the result section we make a discussion according to the literature on the perception and consumer behavior of traditional foods
The references have been expanded with suggested sources.

Round 2
Reviewer 1 Report
Dear authors,
the paper presented with the enhanced text should be revised in a minor sense.
some of the figures take a large space on pages (e. Fig 2,3 ) and then the rest of the page is empty. These pictures are clear and visible so they can be minimised.
In order to enhance the text (lines 420-440) you have put a description what is important for traditional food consumptio.
In conclusion You may link Your findigs according to the set 1,2,3 hypothesis,
while all the text from line 420 to 440 is suitable to be in methodology parte - by hypothesis or in literature review - as You wish.
After these minor changes Your paper will be ready for publishing.
kind regards,
the reviewer.
Author Response
The figures 1, 2 ,3, 4, 5 have been minimized.
All the text from line 420 to 440 was moved to the end of literature review.
In order to enhance the text (lines 420-440) we added a description of the important factors for traditional food consumption.
In conclusions section we made link to the important factors for traditional foods consumption.
Reviewer 2 Report
I appreciate the great effort made by the authors to address my comments and that the manuscript has improved significantly. I still have some observations that can contribute to improving the manuscript. I detail them below.
Although information has been added in the introduction on the importance of traditional foods from a consumer perspective, the rationale for the research is still unclear.
Lines 24 - 29: this information would be better at the end of the introduction, it contributes to the justification of the research.
Line 148: why in the last three decades, if only articles from one decade or period (2012 - 2022) are analyzed?
Lines 188 - 209: It is convenient for the authors to divide the methodology: first write the bibliometric analysis procedures, then describe the survey procedures and finally present the results.
The authors do not describe how the results were analyzed in the methodology section. The results were analyzed with the kruskal Wallis and Mann Whitney tests.
Were tests for normality of the variables performed?
Table 1 gives a total of 279 documents analyzed, and the main text describes an analysis of 251 papers.
In line 271, the authors should use a subtitle to differentiate the results of the bibliometric analysis and the consumer survey.
Lines 278 - 282: this information should be placed in the methodology (survey section).
I believe that the four clusters identified from the bibliometric analysis should be highlighted, and their importance in future research in the context of traditional foods and their implications in terms of public policy should be discussed.
Proofreading of the manuscript is required
Author Response
Dear Reviewer,
Thank you for your comments. In the following you will find our responses and the modifications performed by us in the manuscript following your suggestions.
Q1:Although information has been added in the introduction on the importance of traditional foods from a consumer perspective, the rationale for the research is still unclear.
R1: In introduction section we have introduced paragraphs to clarify the importance of traditional products for consumers and at the same time the purpose of the research within this article.
Q 2 Lines 24 - 29: this information would be better at the end of the introduction, it contributes to the justification of the research.
R2: Following the reviewers suggestion, the lines 24-29 have been moved to the end of introduction
Q3 :Line 148: why in the last three decades, if only articles from one decade or period (2012 - 2022) are analyzed
R3 I made correction, and the “last three decade” was replace with 2012 – 2022.
Q4Lines 188 - 209: It is convenient for the authors to divide the methodology: first write the bibliometric analysis procedures, then describe the survey procedures and finally present the results.
R4: Following the reviewers suggestion, we divided the section “3. Methodology” in 2 subsections: “3.1 Bibliometric analysis methodology”, respectively “3.2 Consumer survey and data analysis methodology”.
Q5:
The authors do not describe how the results were analyzed in the methodology section. The results were analyzed with the Kruskal Wallis and Mann Whitney tests.
Were tests for normality of the variables performed?
R5: We thank the reviewer for this valuable comment. The normality test were performed using the Shapiro-Wilk method. For all the variables the significance levels are lower than 0.05, resulting that all variables used in this study present non-normally distribution. For this reason we decided to use the non-parametric Mann Whitney and Kruskal Wallis tests.
Considering these arguments (Q4 ,Q5 and the following Q8), the Methodology section was reedited
as follows:
- After the row 183 was added the text: “The data for the research were collected using a survey based on a questionnaire.”
- The text in between lines 199-204 was replaced with the following one:
“For the 20 items, the respondents evaluated on a Likert Scale the importance of different preferences and criteria in the decision to choose and consume traditional products. The Likert Scale results were scored as follows: 5 = very important / very satisfied; 4 = important/satisfied; 3 = indifferent (neutral); 2 = less important/slightly satisfied, and 1 = a little important/dissatisfied. Methods of descriptive and inferential statistics were used for analysis of the collected data. The normality of the data was checked using Shapiro-Wilk test. The results show that all significance levels are lower than 0.05, resulting that all variables used in this study present non-normally distribution. Thus, the non-parametric Mann–Whitney and Kruskal–Wallis tests were used to analyze if there were significant differences regarding the residence consumers' perception of the motivational factors in the consumption of traditional products based on socio-demographic characteristics [34]–[36]. Data were analyzed using SPSS Statistics for Windows, version 23.0.”
Q6:Table 1 gives a total of 279 documents analyzed, and the main text describes an analysis of 251 papers.
R6: I made correction of this error.
Q7:In line 271, the authors should use a subtitle to differentiate the results of the bibliometric analysis and the consumer survey.
R7:Following the reviewers suggestion, we divided the section 4. Results in 2 subsections: “4.1 Results on bibliometric analysis”, respectively “4.2 Results on consumer survey”
Q8:Lines 278 - 282: this information should be placed in the methodology (survey section).
R8: We analyzed the reviewer’s suggestion and we concluded that the text between lines 278-282 is well fitted in this position. Still, we understand that the methodologies for interpreting the Likert scales and the resulting data analysis were inadequate presented by us in the methodology section. For this reason, we moved the text from lines 278 – 277 (i.e. previous paragraph to the lines 272-278 suggested by the reviewer) to the Methodology section. This text was adapted to the Methodology section and reedited (see previously response R5)
Q9 I believe that the four clusters identified from the bibliometric analysis should be highlighted, and their importance in future research in the context of traditional foods and their implications in terms of public policy should be discussed.
R9 In the conclusions section, the four clusters identified from the bibliometric analysis was highlighted for the importance in future research in the field of public policy.